# Frailty and Associated Factors among the Elderly in Vietnam: A Cross-Sectional Study

**DOI:** 10.3390/geriatrics7040085

**Published:** 2022-08-20

**Authors:** Trung Quoc Hieu Huynh, Thi Lan Anh Pham, Van Tam Vo, Ha Ngoc The Than, Tan Van Nguyen

**Affiliations:** 1Department of Graduate Training, University of Medicine and Pharmacy at Ho Chi Minh City, Ho Chi Minh City 700000, Vietnam; 2Faculty of Public Health, University of Medicine and Pharmacy at Ho Chi Minh City, Ho Chi Minh City 700000, Vietnam; 3Department of Geriatrics & Gerontology, University of Medicine and Pharmacy at Ho Chi Minh City, Ho Chi Minh City 700000, Vietnam

**Keywords:** aging, frailty, older adults, Vietnam, community, functional decline

## Abstract

Background: Frailty syndrome is common among older people and can lead to various adverse consequences such as falls, cognitive decline, disability, dependent living, increased mortality, excessive drug use, and prolonged hospital stays. Objectives: This research determined the prevalence of frailty and associated factors among older adults in Vietnam. Methods: A cross-sectional study was conducted on 584 older adults across five Ho Chi Minh City wards from November 2020 to January 2021. Based on the modified Fried frailty scale, the participants were divided into three categories: robust, pre-frail, and frail. A chi-square test (or Fisher’s test) examined the relationship between frailty categories and other variables. Multivariable logistic regression used variates with a cut-off of *p* ≤ 0.05 in the univariate analysis. Results: The prevalence rates of frailty and pre-frailty were 19% and 64%, respectively. The most common frailty component was weak grip strength (63.9%), followed by slowness (36.1%), weight loss (21.6%), low physical activity (19.5%), and exhaustion (18.5%). In addition, the prevalence of frailty was significantly associated with age, BMI levels, living alone, and sarcopenia. Conclusion: The community’s prevalence of frailty among older adults is high. Frailty can lead to many adverse consequences for the elderly. As there were some modifiable factors associated with frailty, it should be assessed in older people through community-based healthcare programs for early diagnosis and management.

## 1. Introduction

According to the United Nations, older adults are people who are aged 60 years or older. In recent years, the number of older adults worldwide has increased unprecedentedly. The United Nations estimates that the number of people aged 60 and over is expected to increase by 56%, from 901,000,000 to 1,400,000,000, between 2015 and 2030. The global population is aging rapidly. Currently, 566 million people are ≥65 years old worldwide, with an estimated increase to 1.5 billion by 2050. According to United Nations estimates, over the next 15 years, the number of older adults will increase the fastest in the U.S., Latin America, and the Caribbean (71%), followed by Asia (66%), Africa (64%), Oceania (47%), North America (41%), and Europe (23%) [1].

Vietnam is one of the countries with the fastest population aging in the world. People aged 60 and over accounted for 11.9% of the total population in 2019; by 2050, this number will increase to more than 25%. By 2036, Vietnam will enter a period of an aging population, transitioning from an “aging” society to an “aged” society. This demographic change is expected to occur in Vietnam not only due to a decrease in mortality and an increase in life expectancy but also primarily due to a sharp decrease in birth rates. The declining birth rate in the past decades has significantly impacted Vietnam’s population structure, accelerating the population’s aging rate [2].

Frailty syndrome, a geriatric syndrome, occurs due to the accumulation of multiple functional impairments [3]. The prevalence of frailty in the community ranges from 4% to 59%, while it ranges from 19% to 76% in nursing homes [4]. In Vietnam, The prevalence of frailty in the community ranges from 11.2% to 21.7%, and from 18.5% to 54.9% in hospitalized patients [5,6,7,8,9,10,11]. Since introducing the concept of frailty, gerontologists have studied many assessment tools such as the Fried scale, the Frailty Index (FI), and the PRISMA-7 questionnaire [12]. Among the assessment tools, the Fried scale is the gold standard used in epidemiological studies and predicts clinical outcomes such as re-hospitalization, mortality, falls, and fractures [13,14]. Frailty can lead to many adverse consequences for older adults, such as falls, cognitive decline, disability, dependent living, increased mortality, excessive drug use, and prolonged hospital stays. Moreover, frailty is also an important prognostic sign that can be used to help prevent worsening conditions. Therefore, screening, early detection, and planning intervention for frailty in older people are fundamental in geriatric medicine. However, there have not been many studies on this issue in Vietnam. The purpose of this study was to provide data and contribute to scientific knowledge of the prevalence of frailty and associated factors in community-dwelling older adults in Vietnam by the Fried frailty scale.

## 2. Materials and Method

### 2.1. Sample Size

The sample size was determined using a single population proportion formula:*n* = Z^2^_1-α/2_ × p(1 − p)/d^2^ × DE(1)
with *n* = the required sample size, p = proportion of frail patients, and d = precision (assumed as 0.05), DE (design effect) = 2 (cluster sampling design). Taking references from reports from AT Nguyen (2019) [15], we estimated that the proportion of patients achieving frail would be around 21.7%. Therefore, the sample size for this study was calculated to be at least 524 participants.

### 2.2. Participants

The participants were recruited via health check programs for all older adult patients at the ward health stations, which are homogenous in terms of demographic, social, and geographical characteristics. This is an annual health check-up program for the older adult living in District 9, Ho Chi Minh City. We collaborated randomly with five wards of District 9. Participants were recruited through the persons in charge of community care centers of the ward health station. Therefore, the number of samples was evenly distributed among the wards, with the specific sample number being about 120 participants.

The inclusion criteria were as follows: (1) aged 60 or older; (2) mentally alert and able to listen and give interviews; and (3) able to speak or understand the Vietnamese language.

Participants were not included in this study if they were unable to perform the specific functional test (bedridden people, blind, hearing loss) or unable to communicate (severe cognitive impairment) or those with motor impairment resulting from acute diseases such as recent stroke, coronary heart disease, and cardiomyopathy or with acute musculoskeletal diseases or orthopedic diseases.

### 2.3. Study Design

From November 2020 to January 2021, we conducted a cross-sectional study in District 9, Ho Chi Minh City, Vietnam. Data was collected using interviewer-structured questionnaires on common characteristics (age, gender, education level, and living situation) from participants and performed functional tests [16,17].

The primary exposure variables included the following: Participants’ comorbid conditions were collected based on medical records (chronic diseases, medications, the Charlson Comorbidity Index, current smoking, and BMI levels). Polypharmacy was judged based on daily prescriptions (defined as five or more prescribed medications) [18,19]. In addition, paramedics conducted data that included five well-trained geriatric nurses and nursing students.

Written informed consent was obtained from all participants.

### 2.4. Outcome Assessment

The modified Fried frailty scale [12] determined the level of frailty, including the following five criteria:–Weight loss of more than 5% or 4.5 kg compared to weight in the last 12 months.–Exhaustion: Two questions in the Centre for Epidemiologic Studies Depression Scale were used: “I felt that everything I did was an effort last week” and “I could not get going last week.” People were defined as having exhaustion if they answered “frequently” or “always” at least once [12].–Low physical activity was evaluated using the International Physical Activity Questionnaire—Short Form (IPAQ-SF) [20]. The cut-off values were stratified by sex (women, <270 Kcals/week; men, <383 Kcals/week).–Slowness: We used a 6 m walking time to assess physical performance, with low physical performance defined as a speed below 1 m/s [14,21,22].–Weakness was evaluated based on handgrip strength using a Jamar hydraulic dynamometer (model J00105; Lafayette Instrument, Lafayette, IN, USA). The arm was placed on the side of the body, and a 90° folding elbow held the force meter. The measurements were repeated three times, with the most significant value of the forehand used in the analysis. Weakness was defined as a handgrip strength of less than 14 kg and 28 kg in women and men, respectively [14].

Subjects who met three criteria or more were classified as frail, and those who met from zero to two criteria were classified as non-frail (including pre-frail and robust) [14].

Sarcopenia assessment: We assessed sarcopenia according to the revised version of the Asian Working Group on Sarcopenia 2019 consensus (AWGS), which recommends evaluating muscle strength, muscle quantity, and physical performance while considering ethnic differences [21].

The skeletal muscle mass index is calculated by dividing the appendicular skeletal muscle mass (kg) by the square of the height (m^2^). Appendicular skeletal muscle mass is calculated based on Bioelectrical impedance analysis (BIA) performed using an Inbody device (Inbody 770, multi-frequency segmental body composition analyzer; Inbody, Seoul, Korea).

### 2.5. Statistical Analysis

Frequencies and percentages were used for the categorical variables to describe the participants, and means and standard deviations were used for the continuous variables. Chi-square tests (or Fisher’s tests) examined the relationship between frailty categories and the categorical demographic and exposure variables. In the bivariate analyses, odds ratios (ORs) with a 95% CI were generated using logistic regression. The results were used to evaluate the relationship between frailty status and the characteristics of the participants. Multivariable logistic regression was performed using a stepwise backward selection procedure with a cut-off of *p* ≤ 0.05 in the univariate analysis to identify factors associated with frailty status.

## 3. Result

### 3.1. Demographics and Baseline Characteristics

A total of 584 participants (175 men and 409 women) were eligible for this study. The baseline characteristics of 584 participants are shown in Table 1. The mean age was 69.57 ± 7.25 years, consisting of 55% being aged 60–69, 34.3% being 70–79, and 10.7% being 80 or older. The majority of the participants had normal BMI levels (41.4%). Most participants (74.3%) lived with a relative. Nearly all of them (90.92%) did not smoke. Overall, the rate of participants having sarcopenia was 48.6%, and 5% had severe sarcopenia. The most prevalent comorbidities were hypertension and dyslipidemia. This study also indicated that 47.8% had Charlson Comorbidity Index from 1 to 2 points, and 12.67% of the participants received polypharmacy.

Table 2 shows that among the 584 participants involved in the study, 99 (17%) were robust, 374 (64%) participants pre-frail, and 111 (19%) were frail. The most common frailty component was weakness (63.9%), followed by slowness (36.1), weight loss (21.6%), low physical activity (19.5%), and exhaustion (18.5%).

### 3.2. Factors Associated with Frailty

Table 3 presents the final model from the multivariable analysis. The model revealed that age, BMI levels, living situation, and sarcopenia were predictors of frailty. Participants who were older (aOR = 2.04, 95% aCI = 1.07–391), underweight (aOR = 2.21, 95% aCI = 1.06–4.60), living alone (aOR = 1.64, 95% aCI = 1.04–2.61), or had sarcopenia (aOR = 1.93, 95% aCI = 1.17–3.19) had a higher likelihood of frailty.

## 4. Discussion

Older adults are often at increased risk of adverse health events such as falls, disability, hospital admissions, emergency department visits, and even death. The proportion of older adults hospitalized with a frailty diagnosis is often remarkably high. Accurately identifying patients who may experience adverse consequences is essential for individual care planning and risk assessment for medical treatments or interventions. However, there have not been many studies on this issue in Vietnam. Our study provides more data on the prevalence of frailty in Ho Chi Minh City, Vietnam, a city of the most significant economic center with a high population aging rate. A study of 584 research subjects showed that deficiency is typical among older people. Finally, this suggests potential implications for the development of interventions by which to improve the health of the older adult. In this study, nearly one-fifth of the participants suffered from frailty (19%). Among the diagnostic criteria for deficiency according to the Fried scale, the most common frailty component was weak grip strength (63.9%), followed by slowness (36.1), weight loss (21.6%), low physical activity (19.5%), and exhaustion (18.5%). The rate of frailty was lower than in a previous study in the older adult community (21.7%) [15] and in a hospital *setting* in Vietnam (35.4%) [23]. A possible explanation is that the hospitalized older adult may experience serious health problems, increasing the risk of frailty [24]. The prevalence that we found is comparable to that of the general majority of the older adults in the community (5.4–44%) in other developing countries but is much lower than the number of hospitalized patients (27.8–71.3%) [23]. The results indicate a higher prevalence than in those of other studies in rural communities using similar instruments, such as Colombia (12.2%) [25], Mexico (8.6%), and Turkey (7.4%) [26,27]. This discrepancy may be due to differences in the criteria or items used to assess the degree of frailty. Previous studies have suggested that the frailty rates in each community may vary depending on definition, population, and study design [25,28]. As the older adults in the community have a remarkably high prevalence of frailty, they should be regularly screened for health problems. This can effectively prevent adverse outcomes listed as cardiovascular diseases, depression, fractures, falls, hospitalization, or even death [4,13,29,30].

In line with previous studies, we found that people of an older age are more likely to be frail [15,31]. Moreover, the study also showed a statistically significant association between BMI levels, living situation, and sarcopenia. Participants who were underweight, living alone, and had sarcopenia had a higher likelihood of frailty.

The relationship between frailty and body mass index remains controversial in studies. Some studies have shown an association between underweight and frail patients, whereas others have highlighted a proportional association between obesity and frailty [32]. Changes in body weight are often seen in older people. Body weight tends to increase throughout one’s life expectancy up to 70–80 years, after which body weight gradually decreases. Moreover, malnutrition, being underweight, and obesity are adverse health risk factors in older people. Advanced age has also been linked to changes in body composition, including loss of muscle mass and increased fat mass [33]. In addition, the volume of subcutaneous fat is reduced, while the penetration of lipid into the liver and the organs often increases with age. The increase in overall fat mass and the loss of muscle mass do not depend on weight changes [34].

The situation in which older adults live alone is a social issue of recent concern. In developed countries, approximately one-third of the older adults live alone, which increases with age. In addition, this rate is forecasted to continue to increase over the next 20 years due to the increased life expectancy and improved health status of older adults. Living alone can result in depression and social isolation, which may influence older adults’ functional status [35]. The trend of older adults living alone is increasing in Vietnam. According to the 2019 Midterm Census, older adults living alone or living only with a spouse have increased, while other groups have tended to decline in this respect [36]. Differences between regions in terms of the living arrangements of the older adult are partly due to the impact of migration. There is little scientific evidence of a relationship between frailty and living alone, two common risk factors for adverse events in older adults living in the community. Our research has found an association between these two factors. This finding could provide additional data on the impact of social factors on the health of older adults in Vietnam [37].

The relationship between frailty and sarcopenia has not been fully explored, but both syndromes share many similarities, such as clinical outcomes, pathophysiology, and risk factors [38]. Therefore, many believe sarcopenia to be a component of frailty, but frailty is not a component of sarcopenia [39]. However, there is a significant overlap between the diagnostic criteria used for frailty and sarcopenia. Some studies have classified severe sarcopenia, as defined by the AWGS, as a pre-frail group according to the Fried standards [40]. Sarcopenia is often seen as a predetermined syndrome or a physical criterion for diagnosing frailty.

Our study, in which frailty was assessed using Fried’s criteria, considers a reference standard in frailty diagnosis and contributes to scientific knowledge of the prevalence of frailty and associated factors in the older adult community in Vietnam by the Fried frailty scale. However, this study has several limitations, as follows. Firstly, the cross-sectional nature of this study does not allow conclusions about the predictability of health outcomes of frailty. Second, it was conducted in Vietnam’s urban older adult population, which may not reflect the characteristics of Vietnam’s older adult population. Thirdly, some factors that might be related to frailty were not examined in this study, such as mobility and daily living. Still, the result can be a reference for other community-based studies in Vietnam and provides evidence for screening frailty in the older adult population.

## 5. Conclusions

The community’s prevalence of frailty among older adults is high. Frailty can lead to many adverse consequences for the elderly. As there were some modifiable factors associated with frailty, it should be assessed in older people through community-based healthcare programs for early diagnosis and management.

## Figures and Tables

**Table 1 geriatrics-07-00085-t001:** Participant characteristics (*n* = 584).

Participants’ Characteristics	*n* (%)
Sex	
Male	175 (30)
Female	409 (70)
Age 69.57 ± 7.25 (60–92) *	
60–69 years old	321 (55)
70–79 years old	200 (34.3)
80 years or older	63 (10.7)
Living situation	
Alone	150 (25.7)
With relatives	434 (74.3)
BMI levels	
Underweight (<18.5 kg/m^2^)	49 (8.4)
Normal (18.5–22.9 kg/m^2^)	242 (41.4)
Overweight (23–24.9 kg/m^2^)	117 (20)
Obese (>25 kg/m^2^)	176 (30.1)
Current smoking	
Yes	53 (9.1)
No	531 (90.92)
Sarcopenia	
Non-sarcopenia	271 (46.4)
Sarcopenia	284 (48.6)
Severe sarcopenia	29 (5)
Comorbidity	
Hypertension	306 (52.4)
Dyslipidemia	217 (37.2)
Diabetes	105 (18)
Chronic kidney disease	38 (6.5)
Charlson Comorbidity Index	
0 point	177 (30.3)
1–2 points	279 (47.8)
≥3 points	128 (21.9)
Polypharmacy (≥5 drugs)	74 (12.67)

* Mean ± standard deviation (minimum-maximum).

**Table 2 geriatrics-07-00085-t002:** Prevalence of frailty status and components (*n* = 584).

Characteristics			Frailty Status		
	*n*	%	Robust	Pre-Frail	Frail
Frailty status	584	100	99 (17)	374 (64)	111 (19)
Slowness	211	36.1			
Weakness	373	63.9			
Low physical activity	114	19.5			
Exhaustion	108	18.5			
Weight loss	126	21.6			

**Table 3 geriatrics-07-00085-t003:** Association between frailty status and study sample characteristics (*n* = 584).

Participants’ Characteristics	Frailty Status		Bivariate Analysis			Multivariate Analysis		
	Frail *n* (%)	Non-Frail *n* (%)	OR	95% CI	*p*	aOR	95% aCI	aP
Sex								
Male	27 (15.4)	148 (84.6)	0.71	(0.44–1.14)	0.151	0.66	(0.40–1.10)	0.110
Female	84 (20.5)	325 (79.5)	Ref			Ref		
Age								
60–69 years old	50 (15.6)	271 (84.4)	Ref			Ref		
70–79 years old	40 (20.0)	160 (80.0)	1.36	(0.86–2.15)	0.195	1.09	(0.67–1.77)	0.730
80 years or older	21 (33.3)	42 (66.7)	2.71	(1.48–4.96)	0.001	2.04	(1.07–3.91)	0.032
Living situation								
Alone	40 (26.7)	110 (73.3)	1.86	(1.19–2.89)	0.006	1.64	(1.04–2.61)	0.035
With relatives	71 (16.4)	363 (83.6)	Ref			Ref		
BMI levels								
Underweight (<18.5 kg/m^2^)	16 (32.7)	33 (67.3)	2.24	(1.13–4.440)	0.020	2.21	(1.06–4.6)	0.034
Normal weight (18.5–22.9 kg/m^2^)	43 (17.8)	199 (82.2)	Ref			Ref		
Overweight (23–24.9 kg/m^2^)	22 (18.8)	95 (81.2)	1.07	(0.61–1.89)	0.811	1.22	(0.67–2.22)	0.516
Obese (≥25 kg/m^2^)	30 (17.1)	146 (82.9)	0.95	(0.57–1.59)	0.848	1.18	(0.67–2.09)	0.565
Current smoking								
Yes	6 (11.3)	47 (88.7)	0.52	(0.22–1.24)	0.141			
No	105 (19.8)	426 (80.2)	Ref					
Sarcopenia								
Non-sarcopenia	36 (13.3)	235 (86.7)	Ref			Ref		
Sarcopenia	71 (25.)	213 (75)	2.18	(1.40–3.38)	0.001	1.93	(1.17–3.19)	0.010
Severe sarcopenia	4 (13.8)	25 (86.2)	1.04	(0.34–3.18)	0.939	1.10	(0.34–3.56)	0.876
Charlson Comorbidity Index								
0 point	30 (17)	147 (83)				Ref		
1–2 points	44 (15.8)	235 (84.2)	0.92	(0.55–1.52)	0.043	0.87	(0.51–147)	0.598
≥3 points	37 (28.9)	91 (71.1)	1.99	(1.15–3.45)	0.014	1.73	(0.95–3.16)	0.074
Polypharmacy (≥5 drugs)								
Yes	20 (27)	54 (73.0)	1.71	(0.97–2.99)	0.062			
No	91 (17.8)	419 (82.2)	Ref					

## Data Availability

The study data is available from the corresponding author upon reasonable request.

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
