# Peer review of "Frailty and Associated Factors among the Elderly in Vietnam: A Cross-Sectional Study"

_geriatrics, 2022, doi:10.3390/geriatrics7040085_

Round 1

Reviewer 1 Report

Dear authors

Thank you for the manuscript submission. I agree that more local data is always required to drive better local services. I do have some queries which I hope the authors can help address

- The abstract refers to the prevalance of frailty among the elderly in Vietnam. Not necessarily true as this was a selected cohort of participants in a pre-determined setting. 

- It is not clear reading the manuscript, where participants were recruited from, inpatient vs outpatient, or the type of setting (geriatric medicine wards? ) as this would influence the prevalence significantly. 5 centres were recruitment sites. Were they different or fairly homogenous? Out of interest, were recruitment numbers similar in all 5 sites?

- Line 50: walking speed is not an assessment of frailty. It is embedded into frailty assessment such as Fried

- Recruitment was based on a pre-established sampling scheme. Please elaborate.

- Do help clariy the eligibility criteria. Does mentally alert exclude cognitive impairment? It seems like a very narrow inclusion which would exclude those with cognitive or sensory impairment which are highly prevalent among older people.

- How were the variables for the study decided? Data on mobility and daily living would have allowed analysing the findings in some context.

- The validity and limitation of the study was not discussed. For instance the point listed above which narrows the generalisability of the findings to all older Vietnamese. Or that living with relative might mean due to financial circumstance instead of care needs 

- The term older adults and elderly were used interchangeably. Suggest stick to older adults

Thank you very much. 

Author Response

Response to Reviewer 1 Comments

Point 1: The abstract refers to the prevalance of frailty among the elderly in Vietnam. Not necessarily true as this was a selected cohort of participants in a pre-determined setting.

Response 1:

Thank you for your time reviewing our manuscript and for the valuable suggestions

I've added to the Introduction section:” In Vietnam, The prevalence of frailty in the community range from 11.2% to 21.7%, from 18.5% to 54.9% in hospitalized patients”.

The table below shows some of the research I referenced.

Prevalence

Design

Research

21.7%

Community-dwelling

Nguyen, A.T.; Nguyen, L.H.; Nguyen, T.X.; Nguyen, T.T.H.; Nguyen, H.T.T.; Nguyen, T.N.; Pham, H.Q.; Tran, B.X.; Latkin, C.A.; Ho, C.S.; et al. Frailty Prevalence and Association with Health-Related Quality of Life Impairment among Rural Com-munity-Dwelling Older Adults in Vietnam. Int. J. Environ. Res. Public Health 2019, 16, 3869.

35.4%

Inpatients

Vu, H.T.T.; Nguyen, T.X.; Nguyen, T.N.; Nguyen, A.T.; Cumming, R.; Hilmer, S.; Pham, T. Prevalence of frailty and its associated factors in older hospitalised patients in Vietnam. BMC Geriatr. 2017, 17, 216.

Nguyen, A. T., Nguyen, T. X., Nguyen, T. N., Nguyen, T. H. T., Pham, T., Cumming, R., ... & Vu, H. T. T. (2019). The impact of frailty on prolonged hospitalization and mortality in elderly inpatients in Vietnam: a comparison between the frailty phenotype and the Reported Edmonton Frail Scale. Clinical interventions in aging14, 381.

35.9%

Inpatients

Polypharmacy at discharge in older hospitalised patients in Vietnam and its association with frailty

Thanh Xuan Nguyen1,2 | Tu N. Nguyen1,3 | Anh Trung Nguyen1,2 |

Huong Thi Thu Nguyen1,2 | Thu Thi Hoai Nguyen1,2,4 | Tam Ngoc Nguyen1,2 |

Thang Pham1,2 | Huyen Thi Thanh Vu1,2

11.2%

Community-dwelling

Association of Frailty Status and Functional Disability among Community-Dwelling People Aged 80 and Older in Vietnam

Thu Thi Hoai Nguyen ,1,2,3 Anh Trung Nguyen ,1,2 Thanh-Huyen Thi Vu ,4

Nga Thi Dau ,1 Phong Quy Nguyen ,2 Thanh Xuan Nguyen ,1,2 Tam Ngoc Nguyen ,1,2

Huong Thi Thu Nguyen ,1,2 Thang Pham ,1,2 and Huyen Thi Thanh Vu 1,2

48.1%

Inpatients

Frailty in Older Patients with Acute Coronary Syndrome in Vietnam

Tan Van Nguyen 1,2 Duong Le1,2 Khuong Dang Tran1 Khai Xuan Bui 1 Tu Ngoc Nguyen 3

54.9%

Inpatients

A Pilot Study of the Clinical Frailty Scale to Predict Frailty Transition and Readmission in Older Patients in Vietnam

Tan Van Nguyen 1,2,*, Thuy Thanh Ly 1,3 and Tu Ngoc Nguyen 4

18.5%

Inpatients

Frailty and Adverse Outcomes Among Older Patients Undergoing Gastroenterological Surgery in Vietnam

The Ngoc Ha Than1,2 Thien Nguyen3 Tran To Tran Nguyen 1,4 Tai Pham4

28%

Outpatients

Nguyen, H. T., Nguyen, A. H., & Nguyen, G. T. X. (2022). Prevalence and associated factors of frailty in patients attending rural and urban geriatric clinics. Australasian Journal on Ageing41(2), e122-e130.

12%

Inpatients

Nguyen, T. N., Nguyen, T. N., Nguyen, A. T., Nguyen, H. L., Goldberg, R. J., Nguyen, H. T., ... & Vu, H. T. (2022). Appendicular Lean Mass and Frailty among Geriatric Outpatients. The Journal of Frailty & Aging11(2), 177-181.

2.1%

Outpatients

Khuc, A. H. T., Doan, V. T., Le, T. T., Ngo, T. T., Dinh, N. T., Tran, T. P., & Nguyen, P. H. (2021). Determinants of Frailty among Patients with Type 2 Diabetes In Urban Hospital. Hospital Topics, 1-8.

Point 2: It is not clear reading the manuscript, where participants were recruited from, inpatient vs outpatient, or the type of setting (geriatric medicine wards? ) as this would influence the prevalence significantly. 5 centres were recruitment sites. Were they different or fairly homogenous? Out of interest, were recruitment numbers similar in all 5 sites?

Response 2:

Thank you, we are grateful for all the comments. The research population is the older adults in the community in 5 wards of 1 district in Ho Chi Minh City, which is homogenous in terms of demographic, social, and geographical characteristics. Therefore, the number of samples was evenly distributed among the wards, with the specific sample number being about 120 participants.

Point 3: Line 50: walking speed is not an assessment of frailty. It is embedded into frailty assessment such as Fried.

Response 3: Thank you for your comments. We have removed walking speed in Line 50.

Point 4: Recruitment was based on a pre-established sampling scheme. Please elaborate

Response 4: Thank you very much for your feedback. We randomly selected five wards of the same district; then we made a list to invite all older adults living in that ward to the ward health station at a particular time to choose the research sample.

Point 5: Do help clariy the eligibility criteria. Does mentally alert exclude cognitive impairment? It seems like a very narrow inclusion which would exclude those with cognitive or sensory impairment which are highly prevalent among older people.

Response 5: Thank you very much for noticing that. We exclude cognitive impairment patients.

Point 6: How were the variables for the study decided? Data on mobility and daily living would have allowed analysing the findings in some context

Response 6: Thank you for the comment. This research has not taken into mobility and daily living. That is also a limitation of the study.

Point 7:The validity and limitation of the study was not discussed. For instance the point listed above which narrows the generalisability of the findings to all older Vietnamese. Or that living with relative might mean due to financial circumstance instead of care needs.

Response 7:

Thank you for your comment, I've added the limitations of the study to the Discussion section: Our study, in which frailty was assessed using Fried’s criteria, considered a reference standard in frailty diagnosis, contributes to scientific knowledge of the prevalence of Frailty and associated factors in the elderly community in Vietnam by the Fried frailty scale. However, this study has several limitations, as follows. First, the study didn't find a relationship between frailty and the outcome variables for health listed as quality of life, hospitalization, pathological complications, mortality, and so on. Second, elderly patients requiring emergency status weren't recruited in our study, and there is a possibility that we underestimated the effect of frailty on poor outcomes.”.

Point 8: The term older adults and elderly were used interchangeably. Suggest stick to older adults

Response 8: Thank you, I have edited the manuscript per your comments.

Reviewer 2 Report

“Frailty and Associated Factors among the Elderly in Vietnam: A Cross-Sectional Study”(geriatrics-1836418)

This manuscript aimed to explore the frailty and the associated factors among the elderly in Vietnam. The results revealed that the prevalence rates of frailty and pre-frailty were 19% and 64%, respectively. The most common frailty component was weak grip 20 strength (63.9%), followed by slowness (36.1%), weight loss (21.6%), low physical activity (19.5%), and exhaustion (18.5%). In addition, the prevalence of frailty was significantly associated with age, BMI levels, living alone, and sarcopenia. Overall, this topic is important and this manuscript provides some additional evidence for the frailty among the elderly in Vietnam. However, some concerns appeared after reading the whole manuscript.

1.The first and most important concern is the lack of clearly statement of novelty and motivation to do this research, which partly due to the lack of comprehensive review about the state of art of frailty among the elderly in Vietnam and clearly identified the research gap. Some important papers should be mentioned, such as,

To, T. L., Doan, T. N., Ho, W. C., & Liao, W. C. (2022, May). Prevalence of Frailty among Community-Dwelling Older Adults in Asian Countries: A Systematic Review and Meta-Analysis. In Healthcare (Vol. 10, No. 5, p. 895). Multidisciplinary Digital Publishing Institute.

Than, T. N. H., Nguyen, T., Nguyen, T. T. T., & Pham, T. (2021). Frailty and Adverse Outcomes Among Older Patients Undergoing Gastroenterological Surgery in Vietnam. Journal of Multidisciplinary Healthcare14, 2695.

Nguyen, A. T., Nguyen, T. X., Nguyen, T. N., Nguyen, T. H. T., Pham, T., Cumming, R., ... & Vu, H. T. T. (2019). The impact of frailty on prolonged hospitalization and mortality in elderly inpatients in Vietnam: a comparison between the frailty phenotype and the Reported Edmonton Frail Scale. Clinical interventions in aging14, 381.

Nguyen, H. T., Nguyen, A. H., & Nguyen, G. T. X. (2022). Prevalence and associated factors of frailty in patients attending rural and urban geriatric clinics. Australasian Journal on Ageing41(2), e122-e130.

Nguyen, T. T. H., Nguyen, A. T., Vu, T. H. T., Dau, N. T., Nguyen, P. Q., Nguyen, T. X., ... & Vu, H. T. T. (2021). Association of Frailty Status and Functional Disability among Community-Dwelling People Aged 80 and Older in Vietnam. BioMed Research International2021.

Nguyen, T. N., Nguyen, T. N., Nguyen, A. T., Nguyen, H. L., Goldberg, R. J., Nguyen, H. T., ... & Vu, H. T. (2022). Appendicular Lean Mass and Frailty among Geriatric Outpatients. The Journal of Frailty & Aging11(2), 177-181.

Khuc, A. H. T., Doan, V. T., Le, T. T., Ngo, T. T., Dinh, N. T., Tran, T. P., & Nguyen, P. H. (2021). Determinants of Frailty among Patients with Type 2 Diabetes In Urban Hospital. Hospital Topics, 1-8.

2. The comparison between the current findings and the previous related findings should be thoughtfully discussed in the discussion part.

3. “2.1. Patients” should be 2.1. Participants.

4. Line 184-192 should cite related scientific references.

5. How did you determine the sample size? Did you calculate the sample size needed before formal study?

References:

Lakens, D. (2022). Sample size justification. Collabra: Psychology8(1), 33267.

6. I recommend that the paper be thoroughly proofread and edited for languages and grammars, to enhance readership.

7. I believe the special issue in Healthcare journal would be suitable for this manuscript (https://www.mdpi.com/journal/healthcare/special_issues/Frailty_Community_Dwelling_Older_People)

8. The conclusion should be revised to reflect more insights of this manuscript.

9. Limitations should be added in the discussion part.

Author Response

Response to Reviewer 2 Comments

Point 1: The first and most important concern is the lack of clearly statement of novelty and motivation to do this research, which partly due to the lack of comprehensive review about the state of art of frailty among the elderly in Vietnam and clearly identified the research gap. Some important papers should be mentioned.

Response 1: Thank you for the suggestions – we thank you for providing these references. We have added a section to the introduction on the clear statement of novelty and motivation to do this research. We've also added the citations you suggested.

Point 2: The comparison between the current findings and the previous related findings should be thoughtfully discussed in the discussion part.

Response 2: Thanks you for your comment. We have checked and revised the manuscript.

Point 3: “2.1. Patients” should be “2.1. Participants”.

Response 3: Thank you for the suggestions; we have edited the manuscript as you requested

Point 4: Line 184-192 should cite related scientific references

Response 4: Thank you for the suggestions; we have edited the manuscript as you requested

References

Nam, U. V., & Duc, N. M. (2021). Population ageing and older persons in Viet Nam. General Statistics Office.

Dao, B. T., Tran, N. T., Barysheva, G. A., & Tran, L. S. (2020). Social Security and Population Ageing in Vietnam: A Guarantee for the Elderly People’s Life. International Journal of Criminology and Sociology9, 381-390.

Point 5: How did you determine the sample size? Did you calculate the sample size needed before formal study?

Response 5: The sample size was determined using a single population proportion formula:

with n = the required sample size, p = proportion of frail patients , and d = precision (assumed as 0.05), DE (design effect) = 2 (cluster sampling design) . Taking references from reports from AT Nguyen (2019) , we estimated that the proportion of patients achieving frail would be around 21.7%. Therefore, the sample size for this study was calculated to be at least 524 participants.

References

Nguyen, A. T., Nguyen, L. H., Nguyen, T. X., Nguyen, T. T. H., Nguyen, H. T. T., Nguyen, T. N., ... & Vu, H. T. T. (2019). Frailty prevalence and association with health-related quality of life impairment among rural community-dwelling older adults in Vietnam. International journal of environmental research and public health, 16(20), 3869.

Point 6: I recommend that the paper be thoroughly proofread and edited for languages and grammars, to enhance readership

Response 6: Thank you! We have checked and revised the manuscript. We have used the Language Editing Services from MDPI with a native editor.

Point 7: I believe the special issue in Healthcare journal would be suitable for this manuscript (https://www.mdpi.com/journal/healthcare/special_issues/Frailty_Community_Dwelling_Older_People)

Response 7: Thank you for your recommendation.

Point 8: The conclusion should be revised to reflect more insights of this manuscript.

Response 8: Thank you for your comments. We have update the conclusion: “The community's prevalence of frailty among older adults is high. Frailty can lead to many adverse consequences for the elderly. As there were some modifiable factors associated with frailty. Therefore, frailty should be assessed in older people through community-based healthcare programs for early diagnosis and management”

Point 9: Limitations should be added in the discussion part.

Response 9: Thank you for your suggestions. We have updated our discussion with the limitations of the study: “Our study, in which frailty was assessed using Fried’s criteria, considered a reference standard in frailty diagnosis, contributes to scientific knowledge of the prevalence of Frailty and associated factors in the elderly community in Vietnam by the Fried frailty scale. However, this study has several limitations, as follows. First, the study didn't find a relationship between frailty and the outcome variables for health listed as quality of life, hospitalization, pathological complications, mortality, and so on. Second, elderly patients requiring emergency status weren't recruited in our study, and there is a possibility that we underestimated the effect of frailty on poor outcomes”.  

Round 2

Reviewer 1 Report

None

Author Response

Thank you for your time reviewing our manuscript

Reviewer 2 Report

Thank you for the revisions. Some minor concerns still remained.

1. The determination of sample size should be included in the formal context. 

The sample size was determined using a single population proportion formula:

with n = the required sample size, p = proportion of frail patients , and d = precision (assumed as 0.05), DE (design effect) = 2 (cluster sampling design) . Taking references from reports from AT Nguyen (2019) , we estimated that the proportion of patients achieving frail would be around 21.7%. Therefore, the sample size for this study was calculated to be at least 524 participants.

References

Nguyen, A. T., Nguyen, L. H., Nguyen, T. X., Nguyen, T. T. H., Nguyen, H. T. T., Nguyen, T. N., ... & Vu, H. T. T. (2019). Frailty prevalence and association with health-related quality of life impairment among rural community-dwelling older adults in Vietnam. International journal of environmental research and public health, 16(20), 3869.

2. The first limitation is not appropriate. I believe the nature of the cross-sectional design, that is, no causality can be drawn from cross-sectional study would be a potential limitation.

Author Response

Thank you for your time reviewing our manuscript and for the valuable suggestions.

Point 1: The determination of sample size should be included in the formal context.

Response 1: Thank you for the suggestions. We have updated and revised the manuscript (lines 66-73).

2.1. Sample Size

The sample size was determined using a single population proportion formula:

n= Z21-α/2 x p(1-p)/d2 x DE

with n = the required sample size, p = proportion of frail patients , and d = precision (assumed as 0.05), DE (design effect) = 2 (cluster sampling design) . Taking references from reports from AT Nguyen (2019) [15], we estimated that the proportion of patients achieving frail would be around 21.7%. Therefore, the sample size for this study was calculated to be at least 524 participants.

Point 2: The first limitation is not appropriate. I believe the nature of the cross-sectional design, that is, no causality can be drawn from a cross-sectional study would be a potential limitation.

Response 2: Thank you for your suggestions. We have updated our discussion with the limitations of the study: “Our study, in which frailty was assessed using Fried’s criteria, considered a reference standard in frailty diagnosis, contributes to scientific knowledge of the prevalence of Frailty and associated factors in the older adult community in Vietnam by the Fried frailty scale. However, this study has several limitations, as follows. Firstly, the cross-sectional nature of this study does not allow conclusions about the predictability of health out-comes of frailty. Second, it was conducted in Vietnam's urban older adult population, which may not reflect the characteristics of Viet Nam's older adult population. Thirdly, some factors that might be related to frailty were not examined in this study, such as mo-bility and daily living. Still, the result can be a reference for other community-based stud-ies in Vietnam and provides evidence for screening frailty in the older adult population”.